# End-of-Life and Palliative Care in a Critical Care Setting: The Crucial Role of the Critical Care Pharmacist

**DOI:** 10.3390/pharmacy10050107

**Published:** 2022-08-31

**Authors:** Rhona Sloss, Reena Mehta, Victoria Metaxa

**Affiliations:** 1Pharmacy Department, King’s College Hospital NHS Foundation Trust, London SE5 9RS, UK; 2Department of Critical Care Medicine, King’s College Hospital NHS Foundation Trust, London SE5 9RS, UK; 3Faculty of Life Sciences and Medicine, School of Cancer & Pharmaceutical Sciences, King’s College London, London WC2R 2LS, UK

**Keywords:** critical care pharmacist, end-of-life care, multidisciplinary care

## Abstract

Critical care pharmacists play an important role in ICU patient care, with evidence showing reductions in drug prescribing errors, adverse drug events and costs, as well as improvement in clinical outcomes, such as mortality and length of ICU stay. Caring for critically ill patients around the end of their life is complicated by the acute onset of their illness and the fact that most of them are unable to communicate any distressing symptoms. Critical care pharmacists are an integral part of the ICU team during a patient’s end-of-life care and their multifaceted role includes clinical support for bedside staff, education, and training, as well as assistance with equipment and logistics. In this article, we highlight the important role of the ICU pharmacist using a ‘real-life’ clinical case from our hospital.

## 1. Introduction

Caring for patients at end-of-life is often part of the critical care team’s responsibility since adult intensive care unit (ICU) mortality is reported as approximately 20% [1]. Nonetheless, there are specific challenges unique to the critical care setting which can make the delivery of effective end-of-life care more complex. The acute onset of illness, common in ICU patients, means wishes for end-of-life care may not have been previously discussed with their family. Furthermore, sedated and ventilated patients generally cannot discuss their needs and wishes in such circumstances; this can make decision-making regarding the limitation of life-sustaining treatment particularly challenging [2].

Critical care pharmacists play an important role in ICU patient care, with evidence showing reductions in drug prescribing errors, adverse drug events and costs, as well as improvements in clinical outcomes [3]. Their input in the ICU end-of-life process includes medication choice, doses and titration, and avoidance of potentially devastating errors. Ensuring adequate symptom control and comfort during the dying process are recognised priorities in end-of-life care but the approach to delivering this in the ICU is often different from other settings due to the nature of the environment [4]. The majority of patients transition to end-of-life care while intubated and ventilated, with multiple lines and drains still in situ and often already on high doses of sedatives and analgesics. Given these unique factors, the choice of medication and dose selection in ICU end-of-life care is different from other areas and standardisation is frequently lacking [5]. The involvement of pharmacists in these challenges, although assumed, has not been investigated in detail.

We report a case of a female patient who was admitted to the hospital with refractory status epilepticus and following two weeks of ICU treatment, proceeded to comfort care. Our aim is to explore the pharmaceutical challenges of delivering holistic end-of-life care in a critical care setting and highlight the important role of a pharmacist in the ICU team.

## 2. Detailed Case Description

A 26-year-old female presented to the hospital with recurrent facial twitching episodes which progressed to generalised prolonged tonic-clonic seizures, necessitating critical care admission in order to control them. She had a past medical history of mitochondrial disease, myoclonic epilepsy, and progressive neuro-cognitive decline, followed-up by the neurology department in our hospital. For the last few months, she had been bedbound and was starting to have difficulty with feeding and communication. Her brother had a similar condition and had died in the paediatric ICU of the hospital; she was escorted by her parents who were her full-time carers.

In the ICU, she continued to have recurrent seizures and was started on multiple anti-epileptic agents, with her management being guided by the parent team in collaboration with the intensivists. She was intubated and ventilated due to refractory seizure activity on day five of her admission and underwent extensive examinations, including computed tomography head scans and serial electroencephalograms, which confirmed the progressive nature of the disease. Despite escalation to maximal anti-epileptic treatment, persistent seizure activity was still present on day 17, and given the irreversible nature of the underlying disease, goal-of-treatment conversations were initiated with her parents. They both were aware that the progressive nature of the mitochondrial disease would result in their daughter’s death and accepted the intensivists’ suggestion to move towards comfort care. Their only request was that the intractable tonic-clonic seizures, which they witnessed prior to intubation, were controlled since they found the image of their daughter’s body continuously contracting very distressing.

When the patient was extubated, it quickly became apparent that any weaning of the benzodiazepine and opioid infusions she was on would be impossible, since any dose reduction triggered the status epilepticus. For this reason, the patient remained on the pre-existing morphine (5–8 mg/h) and midazolam (15–20 mg/h) infusions which allowed for spontaneous breathing efforts whilst minimising the seizure activity. Regular glycopyrronium was prescribed for secretion management. A referral to palliative care was requested, as it was thought that discharge to a ward side room would ensure privacy during her last hours/days of life and allow her family to be present. This is a regular practice in our ICU, when a patient is discharged on an end-of-life pathway, so the palliative care team is aware and able to review medication infusions and become involved with family support.

When the palliative care registrar examined the patient, they noted that the doses of opiate and sedative infusions and their intravenous mode of administration were outside of routine ward practice. They raised concerns that these doses might be hastening the time to the patient’s death, a statement that caused distress to the ICU staff who were witnessing the seizures and the family’s upset. A multidisciplinary team (MDT) meeting involving ICU consultants, nurses, pharmacists, and the palliative care team was convened to discuss the best management plan. The pharmacy team presented a small literature review around end-of-life medication dosing which highlighted that, while the doses of sedative and analgesic agents being used to manage this patient were higher than those recommended in clinical practice guidelines or which may be used in settings outside the ICU, they were in line with doses reported in the literature in similar settings, and, crucially, there was no evidence to suggest that such doses hastened death. This provided clarity and reassurance to the MDT that it was appropriate and safe to continue with the medication doses being administered to effectively manage this patient’s symptoms while she was cared for in the ICU. Pharmacists undertook education and support of the clinical staff that looked after the patient to ensure a good understanding of the agreed medication management plan. She remained in the ICU, where the necessary doses of sedative and analgesic infusions were routinely reviewed and titrated as per the patient’s symptoms. The patient passed away 56 h after the transition from active treatment to comfort care, seizure-free and with her parents present.

## 3. Discussion

### 3.1. ICU End-of-Life Care Medications and Doses

Pharmacotherapy is considered the cornerstone of symptom control in end-of-life care, with opioids being the most commonly used agents due to their ability to manage pain, anxiety, and dyspnoea. Morphine use is reported in 60% of patients, while benzodiazepines are used in 45–82% of patients in ICU near the end-of-life. Concerns have been raised as to whether the use of such agents may hasten death due to over-medication [4], however, data have been inconsistent [6]. The majority of studies have shown no association between doses of sedation and analgesia and the shortening or delay of death [7,8,9], with the study by Mazer [10] reporting that higher doses of opioids given during the last hour of life after terminal extubation correlated with a statistically significant longer time to death, and Long et al. demonstrating that higher doses of opioids and benzodiazepines were associated with a shorter time to death [11]. The potential for moral conflict and the risk of offering suboptimal symptom control during the end of life for fear of potentially hastening death has been addressed by ethicists with the use of the Doctrine of Double Effect, by which an action is ethically justifiable if its intention was morally good, even there is a foreseeable but unintended morally bad side effect [12].

One of the main challenges in ensuring appropriate medication doses in the ICU is that many patients have impaired cognition or communication ability and may not be able to accurately report their level of discomfort. Literature suggests that, despite sedation aimed at relieving symptoms, delirium and dyspnoea remained an issue, whereas approximately 50% of critically ill patients reported being in pain during the last three days of life [13]. Furthermore, since patients often transition to end-of-life care after withholding life-sustaining treatment, rather than withdrawing it [6], they are still receiving invasive, potentially uncomfortable, interventions while in their last days/hours of life. It is likely due to the need for the ICU staff to ensure that the patient is comfortable in the time leading to their death that medication is not weaned during the transition period. Two recent systematic reviews [5,6] found that the mean doses of analgesic and sedative agents before and after withholding/withdrawal of life support remained consistent through time, albeit higher than those proposed in a recent consensus statement for ICU end-of-life patients [13]. Concordant to the previously mentioned studies [7,8,9], a recent Cochrane review found no statistically significant difference in the time from admission to death between sedated and non-sedated groups, suggesting that end-of-life drugs did not hasten death [14]. Data from our institution are consistent with published literature and confirm the higher doses administered (Table 1).

The literature on medication dosing during end-of-life care in the critical care setting remains limited, however, there are a number of good quality studies available which provide reliable data to support decision making. There are several small observational studies, including approximately 75 patients [7,10], and larger studies, including a meta-analysis [5] involving 13 studies with a total of 2684 patients, as well as a large prospective study [8] which took place across 37 ICUs in 17 European countries, recruiting 4248 patients. Further large-scale studies would be beneficial in providing a greater consensus on the most appropriate treatment strategies in this area.

### 3.2. Role of the Critical Care Pharmacist

Critical care pharmacists are an integral part of the critical care MDT, a fact that has been highlighted in a large amount of literature, ‘arguably more than which exists for any other clinical team member in the ICU and likely more than some of the standard-of-care interventions (such as mechanical ventilation)’ [3]. The indication is clear, the presence of pharmacists is associated with improved care, translated into reduced mortality, ICU length of stay, and adverse drug events (either preventable or non-preventable) [15]. The ‘standard’ role of the ICU pharmacist includes the provision of medication information, clarifying and correcting medication orders, identifying drug interactions, thus avoiding potential adverse drug events, and recommending alternative therapies [15]. Nonetheless, a multitude of other roles have emerged, following the rapid progression in the field of pharmacotherapy, the increasing complexity, and specialisation of critical care medicine, and the realisation that multidisciplinary care leads to improved outcomes. A recent multi-organisational position paper on critical care pharmacy services included 82 recommendation statements addressing various domains of critical care pharmacy practice [16].

Focusing on the challenging area of ICU end-of-life care, pharmacists have a crucial role to play in ensuring patients’ symptoms are managed safely and effectively. While critical care pharmacists are not specialists in palliative care, they are experts in managing medications at the end-of-life in the ICU environment which is very different from other settings where palliative care may be delivered, advising on monitoring and dose titration to manage the vast array of symptoms that ICU patients may suffer, critical care pharmacists prioritise patient comfort and safety at all times.

Many patients will remain in ICU during end-of-life care with intravenous access in situ, however, most standardised guidance on medication dosing is via the subcutaneous route, with little guidance on intravenous dosing. Pharmacists are best placed to support medication guideline development, addressing the specifics of ICU management; examples would be the development of equivalence tables for analgesic and sedative medications via different routes, the review of maximum dosing adjusted for the complexity of ICU patients, and the assessment of potential drug interactions due to the known polypharmacy in these patients. Ensuring that transitions between different routes of administration (intravenous, subcutaneous, and oral) are done seamlessly, critical care pharmacists share the responsibility of patient care with other ICU staff, allowing intensivists to dedicate more time to complex decision-making around the end of life, family support, and avoidance of conflict. In our institution, critical care pharmacists worked alongside nursing and medical consultant colleagues to develop guidelines for intravenous anticipatory medications as well as adapting existing Trust guidance on syringe driver management for the ICU, both of which have been extremely valuable.

There is a significant role for pharmacy in the education and training of critical care staff, especially nurses and trainee doctors, on end-of-life medication management. On a practical basis, running a subcutaneous syringe driver containing multiple drugs at a constant rate is unfamiliar to many ICU nurses, who are used to titrating an infusion rate according to observations. Reinforcing the learning around the complexity of this practice and the different approach to dose titration when co-administering multiple drugs will minimise the risk of medication errors during a very challenging time for patients and families. Keeping up to date with the newest developments around drugs for symptom-relief, allows ICU pharmacists to become experts in the end-of-life field, making recommendations during ward rounds and providing invaluable bedside teaching. Furthermore, clarifying the difference between the administration of opioids and benzodiazepines for sedation vs. comfort around end-of-life, and providing information on the doses required for the former vs. the latter is paramount in order to avoid both patient over-sedation and potential moral conflict for healthcare professionals [6]. Lastly, the review of patient drug charts, which is a standard activity for a critical care pharmacist, is even more important when a patient’s care focuses purely on comfort. Assisting the multidisciplinary team with the discontinuation of medications that no longer serve the goals of care, while adding new ones in anticipation of end-of-life symptoms, not only ensures optimal management during the final period of life but also has positive cost and drug-stewardship implications. Having a permanent forum in nursing and medical teaching fora consolidates the role of pharmacy as a source of expertise and support for all members of the clinical ICU team.

Electronic prescribing system optimisation is another area where pharmacy input is proving to be invaluable; assisting with the design of standardised order sets and their configuration for the dynamic physiology and rapidly evolving disease states of critically ill adults is slowly becoming an integral part of the pharmacist’s role in the new, digital, paper-free ICU. Specifically, around the complex end-of-life period, monitoring patient symptoms and guiding drug therapy in close collaboration with bedside staff contributes to true multidisciplinary care and the early recognition of and prompt response to any signs of discomfort. The proactive pharmacy review of real-time dose changes in sedative infusions, syringe drivers, and ‘as required’ doses of anticipatory medications also allows for good prescription stewardship with clear indications and instructions documented. Such safe practices have been shown to minimise the risk of patient harm [3] and reduce the unnecessary use of resources [15]. Additionally, the presence of specialist pharmacists in ICU ward rounds is integral to ensuring that, for patients discharged from critical care on an end-of-life pathway, the transfer of care process is done seamlessly. The transcription of anticipatory medication from the ICU to ward drug charts (which potentially can be on different informatics systems) ensures holistic comfort care continues following discharge. As stated before, pharmacists often support the conversion of medication regimens from the intravenous to the subcutaneous route, with appropriate dose adjustment, and communicate with ward pharmacists to hand over the current plan and any medication issues for follow-up. ICU pharmacists also work closely with specialist palliative care teams when discharging complex patients receiving end-of-life care from the ICU in order to ensure current medication dosage regimens and monitoring requirements can be delivered safely in a non-ICU environment or to adjust these appropriately where necessary.

However, while pharmacists in the ICU have proven to have a positive impact not only on patients but also on the entire team, their presence in critical care is inconsistent: not all ICUs have specialist pharmacists but even in those that do, their presence on the daily ward rounds is not guaranteed [3]. Financial constraints, the slow uptake of the evidence demonstrating the value of ICU pharmacists, and the absence of robust cost-benefit analyses are some of the reasons proposed to explain the inconsistency [3,15]. Despite several studies showing cost-effective care via a reduction in preventable adverse drug events, no specific economic evaluations and comparative cost analyses are available for critical care, unlike for the ward setting [17]. Furthermore, specialist critical care training is not part of many pharmacy training curricula, a fact that restricts the number of available pharmacists, who are sufficiently experienced to start working in an ICU without previous ‘on-the-job’ training.

## 4. Conclusions

In summary, there are several factors unique to the critical care environment which make the delivery of end-of-life care challenging. However, the participation of specialist ICU pharmacists in the multidisciplinary critical care team ensures that patient outcomes, such as mortality, length of stay, and adverse medication errors, are reduced. Their skills, knowledge, and expertise allow them to monitor and manage patients effectively, often using medications, doses, and routes that may not be appropriate outside the ICU setting. Apart from their contribution to clinical care, ICU pharmacists work collaboratively with the critical care team to deliver safe and effective end-of-life care through guideline development, education activities, informatics optimisation, and clinical review.

## Figures and Tables

**Table 1 pharmacy-10-00107-t001:** Mean medication doses used in critical care during end-of-life care—comparison between a recent systematic review, clinical practice guidelines, and audit data. * Doses calculated for a 70 kg patient.

		Systematic Review	Clinical Practice Guidelines [6]	King’s College Hospital Audit Data
**Mean Medication Doses**	Morphine	6–17.6 mg/h	2–7 mg/h	2 mg/h
	Fentanyl	100–303 mcg/h	35–140 mcg/h *	70 mcg/h
	Midazolam	7.4–13.8 mg/h	1–5 mg/h	1.7 mg/h

## Data Availability

Not applicable.

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
