# Peer review of "End-of-Life and Palliative Care in a Critical Care Setting: The Crucial Role of the Critical Care Pharmacist"

_pharmacy, 2022, doi:10.3390/pharmacy10050107_

Round 1

Reviewer 1 Report

This is a case report that discusses the important role of an ICU pharmacist in the end of life setting. The authors describe a case of a young woman admitted to the ICU with intractable seizures due to an incurable disease that ultimately results in the decision to change to comfort focused care. The authors describe the role ICU pharmacists can play in ensuring excellent clinical care to ensure patients receive as much comfort as possible. They address the controversy that over sedation or analgesia may result in hastened death and provide important data to support the potential harm in not treating symptoms aggressively enough. The authors also appropriately describe the role of pharmacists as educators either directly teaching other staff and trainees or indirectly through affecting order sets. They provide important arguments for the incorporation of an ICU pharmacist in all multidisciplinary teams.

I believe this report would be strengthened further if the authors could address the pharmacists’ expected training in end of life care. Similar to other healthcare providers, not all pharmacists will be trained or experienced with specifically end of life medication management or adjustments. Do they expect specific training in palliative care? Do their pharmacists join in family meetings to understand medication recommendations in the context of the goals of care? Could they further discuss the role of a pharmacist specialized in palliative care and how ICU pharmacists work together with palliative pharmacists, when teams have these specialists available?

Lastly, I believe the authors could improve their discussion by addressing the impact of palliative are specialists on end of life medication management. How do ICU pharmacists without palliative care training enhance the educational gaps and clinical outcomes when a palliative care team is involved?

Author Response

We are truly grateful to the reviewer for their insightful comments. We completely agree on the importance of highlighting the difference in experience and training of ICU pharmacists and palliative care specialists and how the two can work together in the management of patients at end-of-life. We have inserted the requested sentences, which clarify the training of ICU pharmacists in end-of-life care and their area of expertise, as well as highlighting when and how ICU pharmacists would work with palliative specialists on end-of-life medication management.

Reviewer 2 Report

This is an interesting and well written article, which provides an insight into a pharmaceutical challenges with delivering holistic end-of-life care in a critical care setting and highlight the important role of a pharmacist in the ICU team.

I recommend only minor revision and text editing.

Detailed Case Description

The information on the final pharmacists’ recommendations in the presented case is missing (changes in the drug therapy). The useful information for the readers would be to provide details on the pharmacists’ intervention, such as the proposed dose of morphine and midazolam in the presented case and the acceptance of the intervention and/or final decision of the team on the ICU end-of life care medications and doses.

Likewise, more details regarding lines 88-89 would be useful (Pharmacy presented a small literature review around end-of-life medication dosing and undertook the education and support of the clinical staff that looked after the patient. What was the outcome of the literature review and the final decision regarding the presented patient?

line 86 please provide the full term of the abbreviation MDT

Discussion

lines 103-104 -could you please verify if this is correct: “with the study by Mazer 10 reporting that higher doses of opioids given the hour before death after terminal extubation correlated with statistically significant longer time to death…”. It is confusing that time to the death was prolonged, while opioids were given HOUR BEFORE DEATH.

lines 126-128 – the authors refer to the data from “their institution which are not presented in the article”. This leaves some questions to the readers and. Could you please provide more information on the mentioned data and discuss it further. This will contribute the understanding of the text. Otherwise, I would suggest to exclude this part.

I suggest adding a paragraph in the discussion section with concise critical appraisal of the literature you refer to (in section part 3.1.), such as what type of studies are available and what is the sample size. This will provide important information to the reader to answer the question if the available literature provides the necessary information or do we need more evidence.

References

Please edit the style of the references in order to meet the journal criteria. For example, year of the publication is written in different format for different references.

Author Response

  • The reviewer requests more information on the pharmacist’s intervention on the patient case and the outcome of the literature review on final treatment decisions.

We thank the reviewer for the opportunity to provide more important details on the contribution of the pharmacist to the patient case, which is now included.

  • The reviewer also suggests adding a concise critical appraisal of the available literature to allow readers to determine if existing evidence is sufficient. 

We acknowledge the importance of adding this information to the manuscript. We have inserted a small paragraph critically appraising the existing literature around end-of-life medication management in critical care, as requested.

  • The reviewer suggests some changes to abbreviations (e.g. MDT), requests a review of the way the findings of the study by Mazer are described and suggests correction of the referencing format in line with the journal style.

All of these comments are gratefully received and have been acknowledged and updated accordingly in the manuscript. We hope our message is now able to be more clearly understood.

Reviewer 3 Report

Abstract is concise and has given required gist of the article. This case report is comprehensive with detailed description of role and responsibilities of pharmacists in ICU patient care. Case description proved the pharmacist’s involvement in dose titrations of patient’s opioids and analgesics on an end-of-life pathway. Also there is clear explanation of  the process of comfort care organized palliative care team with the help pharmacist. Adequate discussion with highlighting the need and importance of ICU pharmacists in delivery of end-of-life care challenging.

1. What is the main question addressed by the research?

Authors specifically tried to explain the problem involved with use of opioid and analgesics use in End-of -life patients.  This case report describes the essence of dose titrations and standardization of safe/effective doses of opioid and analgesics, how the pharmacists play crucial role with analyzing different factors such as studying patient charts and disease stage, etc.

2. Do you consider the topic original or relevant in the field? Does it
address a specific gap in the field?

Yes, definitely it is relevant in the field. As a pharmacist, I agree with authors point of view and they explained in detail the current existing gap with involvement of pharmacist in ICU patient care, also supported their argument with appropriate case description the pros of having pharmacist review patient chart and also later in the discussion they showed cons involved due to lack pharmacist staff in ICU.

3. What does it add to the subject area compared with other published
material?

When compared to other publications, in this case report authors presented both advantages and way the pharmacists perform analysis of the patient condition to provide safe/effective doses for the patients that require palliative care.

For example, clarence et al., in their manuscript “Do we need a pharmacist in ICU?” They only discussed about how pharmacist resolve drug -related problems in ICU and about how the hospitals have financial difficulties to appoint pharmacist in ICU.

In this current report, author further explained other factors involved like Opioids and analgesics (morphine and benzodiazepines), about development of equivalence tables via different routes for dose adjustments. Also, they provided concept of syringe driver management.

4. What specific improvements should the authors consider regarding the
methodology? What further controls should be considered?

It would be better if authors could provide the process of dose titration that pharmacist done in this case with 26-year-old female. For example, if they could briefly discuss the findings of literature review regarding dosing information which pharmacist found out.

5. Are the conclusions consistent with the evidence and arguments
presented and do they address the main question posed?

Yes, in the conclusion the advantages of having pharmacists for patient care and the importance of their skills for effective patient safety/care in ICU addressed the question.

6. Are the references appropriate?

Yes, adequate references provided.

7. Please include any additional comments on the tables and figures.

None

8. Is there any definition that is not clear?

 None

Author Response

The reviewer suggests that the manuscript would be improved by adding more information on the findings of the pharmacist’s literature review in the patient case and subsequent dosing information provided.

We are most grateful for the opportunity to provide more information on this and have added sentences detailing what the pharmacist’s literature review showed and how this impacted on the subsequent decisions made by the multidisciplinary team on medication management for this patient. We hope this addresses these comments satisfactorily.